# Harsh Physical Discipline and Externalizing Behaviors in Children: A Systematic Review

**DOI:** 10.3390/ijerph192114385

**Published:** 2022-11-03

**Authors:** Marthe Wiggers, Fred Paas

**Affiliations:** 1Department of Psychology, Education & Child Studies, Erasmus University Rotterdam, 3062 PA Rotterdam, The Netherlands; 2School of Education/Early Start, University of Wollongong, Keiraville, NSW 2522, Australia

**Keywords:** physical punishment, externalizing behaviors, parenting, parental warmth, culture, education

## Abstract

There is growing debate in the parenting literature as to whether using physical punishment to discipline children is an effective strategy or leads to the development of aggressive behaviors and other antisocial attributes. The aim of the current literature review is to examine the association between harsh physical discipline and the development of externalizing behaviors in children, as well as the suggested moderators of this relationship. Secondly, the findings regarding the effects of harsh physical discipline on children’s educational outcomes are reviewed. Articles were selected from relevant databases while maintaining an inclusion and exclusion criteria, with a total of 22 articles included in this review. Strong associations between parental corporal punishment and a range of child behaviors were indicated by the literature, and cultural normativeness was implicated as a moderator of these effects. Results regarding the role of parental warmth as a moderator did not provide a firm conclusion. Finally, the findings suggest that when a child is subjected to physical discipline in the home, their life at school may be adversely affected by impaired cognitive performance, peer isolation, and behavioral problems. The primary limitation of the studies reviewed is the use of self-report data and correlational analyses, ruling out the possibility of inferring causal relations. Nonetheless, the results indicate the necessity of encouraging parents and caregivers to avoid physical punishment as a disciplinary tactic while providing them with the tools to explore alternative practices.

## 1. Introduction

Physically punishing a misbehaving child is an ordinary part of parental discipline across continents [1]. In many cultures, hitting, spanking, or slapping a child is nothing out of the ordinary, despite the large body of research claiming that physical punishment is harmful, ineffective, and associated with increased odds of childhood maltreatment [2,3]. Approximately 63% of children are regularly subjected to harsh physical discipline by parents or caregivers, even though The United Nations Committee on the Rights of the Child has stated that physical punishment is a form of violence that threatens a child’s security, and several countries have laws against the practice [4].

A thorough understanding of how physical punishment affects children is not yet available, as some studies suggest physical punishment is associated with neutral or positive outcomes, while other research has shown that the use of physical punishment is linked to a range of negative child and adolescent outcomes [5]. Childhood and adolescent externalizing behaviors are by far the most studied of these outcomes, and research suggests that the use of harsh physical punishment in children is associated with increases in externalizing behaviors over time [3]. This classification of behaviors can be characterized by outward expressions that reflect negatively on a child’s external environment and can be a major risk factor for aggression, delinquency, antisocial behavior and conduct problems. The social learning theory, which posits that parental use of physical discipline teaches children that violence is sometimes effective or necessary, is one approach to explain this phenomenon [6]. Children observe and imitate the behaviors of those around them, especially those they consider to be role models (e.g., their primary caregiver), as highlighted, for example, by a child’s increased likelihood of exhibiting aggressive behavior once they have been subjected to harsh physical punishment themselves [7]. Further, while corporal punishment has been shown to secure immediate compliance and help a parent gain control of a child in the moment, it is not unlikely that instances of physical punishment can escalate into more serious maltreatment. For example, in a study by Afifi et al. [2], harsh physical punishment was associated with increased odds of childhood maltreatment, including sexual, physical, and emotional abuse. These findings indicate that reducing children’s exposure to physical punishment is an effective way to reduce exposure to more severe forms of maltreatment as well.

Ultimately, growing up in a well-balanced environment fosters positive developmental outcomes, and disruption in the safety and stability of a child’s home life may lead to disruption in other domains as well [4]. In the classroom, corporal punishment of a child may worsen their problematic and disruptive attributes, making it difficult for them to succeed academically. For example, corporal punishment has been linked with declines in school engagement as students may be less focused on learning while experiencing heightened levels of stress as a result of the exposure to physical punishment [8]. Another vital part of success in the traditional classroom environment relates to the process of building successful and positive relationships with peers [7]. Effective socialization requires that a child internalizes moral and social norms as he or she learns how to communicate effectively. Unfortunately, studies have shown that this process is disrupted when children are physically punished as the reasons for behaving appropriately are not explained to them [1]. As a result, a child may be rejected by peers as they struggle to effectively interact with others, making their life at school increasingly difficult. This further highlights the importance of encouraging parents to consider alternative methods of discipline since physical punishment at home has spillover effects into other domains and can seriously threaten a child’s potential for success in later life [1,3].

At first glance, the advice is clear; physical discipline is unnecessary, ineffective, and detrimental to a child’s developmental outcomes. However, psychologists have debated the topic for decades and failed to reach a clear consensus due to various limitations [1]. Although the association between harsh discipline and externalizing behaviors is well-established in the literature, a major limitation in the field is that most study samples are White, middle-class families in the United States [3]. Recent debate in the literature has shown that there are ethnic and cultural differences which may moderate the effects of physical discipline on a child’s developmental outcomes. For example, African American parents have been shown to endorse physical punishment as effective and acceptable more strongly than European American parents [9]. As a result, in an environment where physical punishment is normative, a child may not receive the message that their parent is acting out of control and may not react to the behavior as strongly [7]. Further, a new perspective on physical punishment has emerged in recent years, namely the conditional spanking perspective. According to this perspective, there are conditions under which spanking or other forms of physical discipline can be effective, and the context and implementation of these practices play a large role in the way these tactics are interpreted. For example, in their meta-analysis, Larzelere and Kuhn [10] found that conditional spanking was more strongly associated with reductions in non-compliance or antisocial behavior than alternative disciplinary tactics. In their study, physical punishment was only associated with detrimental effects when it was used as the predominant disciplinary method or when its usage was severe. Gershoff [1] also found parental corporal punishment to be positively associated with immediate compliance, which suggests there are situations in which this kind of discipline may be favorable. Nonetheless, this may not achieve children’s compliance and the internalization of these morals and social standards in the long-term. Of course, it is highly plausible that the same mechanisms do not apply across populations and that the assessment of corporal punishment differs between ethnic and cultural groups. More recently, religion has also emerged as an important marker of the cultural differences in parent’s use of corporal punishment [11]. To this end, it is vital for researchers within the field to consider the wider cultural context when investigating these relationships.

Another key limitation in the research is that there is inconsistency in the way harsh physical punishment is defined or characterized by both researchers and participants. As a result, the line between child abuse and corporal punishment might be unclear as parents may be unwilling to admit to physically disciplining their child or unaware that their behavior might be considered abusive, placing the reliability and generalizability of the findings into question [1]. As a result of these limitations, policy makers may not be fully aware of the deleterious effects of physically punishing a child as the data does not accurately convey what is happening behind closed doors. Finally, there are several moderators discussed in the literature which have been indicated as buffers of effects of harsh physical punishment on childhood aggressive behaviors and other conduct problems. Parental warmth, characterized by affection, comfort, concern, and nurturance, has gained particular attention as it encourages trust between the parent and child, promotes reciprocity, and encourages the regulation of positive and negative emotions [12]. These findings have important implications as they suggest that parents can, at least in part, compensate for harsh discipline by showing high levels of warmth toward their child [13]. Clearly, a comprehensive review in the field is highly necessary to obtain an extensive understanding of the aforementioned effects. 

To this end, the current paper will review the literature and seeks to answer the following questions: what are the effects of harsh physical punishment of children on the development of externalizing behaviors, and which factors moderate the effects of this punishment on the development of externalizing behaviors? More specifically, this review will focus on the aggressive behaviors, antisocial behaviors, and behavioral adjustment problems that have been linked with the use of corporal punishment to discipline children. Additionally, educational outcomes will be briefly discussed as these highlight the harmful short- and long-term effects of physical punishment on a child’s ability to succeed in everyday settings. To investigate these effects, peer-reviewed studies in both familial and classroom environments will be reviewed and critically evaluated. Only studies examining physical punishment of a child as a disciplinary measure will be included; studies of child abuse or maltreatment will be excluded as these fall outside the scope of this review. The definition provided by Gershoff [1] will be used to guide this distinction; behavior that does not result in physical injury (for example, spanking or slapping) is not considered child abuse, while punching or kicking can be considered abusive as it risks injury to the child. This paper is theoretically and practically relevant as the abundance of contradictory literature makes it difficult to optimally advise the practitioners, policy makers, and healthcare professionals who parents and caregivers may consult for advice regarding effective disciplinary methods.

## 2. Materials and Methods

For this systematic literature review, sources were screened and selected during April, May, and June of 2022 on several databases, including SCOPUS, Web of Science, and PsycInfo. The key words and subjects included in the search strategy were as follows: harsh physical punishment, corporal punishment, harsh discipline, externalizing behaviors, aggression, conduct disorder, behavioral problems, adjustment problems, and antisocial behavior. The search terms mentioned previously were combined with terms relating to the effects of corporal punishment of the child on the adjustment of children in the classroom and their subsequent academic achievement. Search terms for this area of interest included: classroom performance, academic performance, school adjustment, and academic achievement. Boolean techniques were used to focus the research and search for results more appropriate to the research question(s). Finally, asterisks were used in combination with specific search terms to broaden the results; for example, agress * showed results with the words aggression and aggressive. The specific search query used for this literature review was as follows:

(TITLE-ABS-KEY (“harsh physical punishment” OR “harsh punishment” OR “physical punishment” OR “harsh discipline” OR “physical discipline” OR “corporal punishment”) AND (“antisocial behavior” OR “conduct disorder” OR “behavior* problems” OR “externalizing behaviors” OR agress *)) OR TITLE-ABS-KEY ((“harsh physical punishment” OR “harsh punishment” OR “physical punishment” OR “harsh discipline” OR “physical discipline” OR “corporal punishment”) AND (“classroom performance” OR “school adjustment” OR “academic performance” OR “academic achievement”) AND NOT TITLE-ABS-KEY ((“abuse” OR “sexual abuse” OR “physical abuse”))) AND (LIMIT-TO (LANGUAGE, “English”)).

### Systematic Literature Search

An overview of the literature search process is presented in the PRISMA flow diagram in Figure 1 [14]. In the initial stage, articles were selected if the keywords mentioned above (or combinations thereof) appeared in the title, abstract, or keywords of the paper. Following this, additional screening was conducted to assess the quality of the articles. To do so, the scientific journals from which the articles were derived were evaluated along with the number of times the article had been cited. Articles were only included if they were peer-reviewed and written in the English language. Literature reviews were excluded from the analysis. When articles were deemed appropriate and relevant, the introduction, method, and discussion were screened (with emphasis on the introduction and discussion) to assess whether they answered one of the research questions mentioned above. The snowball method was also used by looking at the reference lists of the articles that discussed the relevant subjects. For the inclusion criteria to be met, only empirical articles discussing the impact of harsh physical punishment by a parent or caregiver on a child’s externalizing behavior or educational outcomes (or the moderators of these effects) were included. Due to the nature of the subject area, it was unavoidable that most studies relied on teacher, caregiver, or self-reported questionnaires. An exclusion criterion was also followed during the search process. Articles were excluded if they discussed serious issues of maltreatment or abuse, rather than physical punishment, as a disciplinary method. Articles were further excluded if the population of the study was composed of adults, as this falls outside the scope of the current research. Moreover, non-English articles were excluded, as well as studies that assessed physical punishment inflicted by a teacher or other school staff member rather than a parent or caregiver. Finally, after reading the entirety of the selected papers, any article that did not explicitly answer the research question was excluded from the review. An overview of the included articles is presented in Table 1. 

## 3. Results

The link between harsh physical discipline of children and the development of externalizing behaviors has been well-established in the literature [1]. It is important, however, to acknowledge that most of the research in the field relies on correlational data, thus ruling out the possibility of establishing a causal relationship or a determining temporal sequence [15]. Furthermore, assigning parents to a punish or no-punish intervention group is not plausible, thus most of the findings of physical discipline research come from questionnaires or recollections of experiences of corporal punishment [1]. Nonetheless, parenting research has provided valuable insight while addressing some of these methodological concerns; for example, by controlling for parental stress levels, socio-economic factors, and further child and parent confounds [15]. The following section of this review will critically evaluate the parenting literature pertaining to physical discipline effects on children’s externalizing behaviors and the suggested moderators of these effects, namely cultural normativeness and parental warmth. Finally, the associations between harsh physical discipline and young children’s educational outcomes will be discussed to highlight the implications of behavioral problems in domains outside of the family environment.

### 3.1. Physical Discipline and Externalizing Behaviors in Children

One of the most common methods of parental physical discipline is spanking. Spanking can be characterized as a slap with an open hand, especially on the buttocks, often used to correct undesirable behavior in children [16]. Unfortunately, studies have found that spanking can actually have the opposite effect [17]. Literature points towards a link between spanking and higher levels of aggression in children, as well as children’s endorsement of spanking as an effective conflict solution method [18]. In their longitudinal study, Mackenzie et al. [17] examined the prevalence of maternal and paternal spanking of children at ages three and five. Rule-breaking, aggression, and destructive habits of the child were measured using the Child Behavior Checklist (CBCL), a commonly administered questionnaire to assess behavioral and emotional problems [19]. Interestingly, the results of their research indicate that maternal spanking administered at high and low frequency at age five was a significant predictor of higher externalizing behaviors at age nine. These findings go against the argument that normative harsh discipline is not detrimental, suggesting even low-frequency spanking can have a negative effect on a child’s behavioral adjustment. In a similar study, Mulvaney and Mebert [20] utilized the CBCL and observations of the family environment to examine the impact of corporal punishment (CP) on children’s behavioral problems. This study contained a younger sample, assessing children at 36 months and first grade, and found that CP contributed to negative behavioral adjustment at both time points. A particular strength of this study was that the assumed causal variable (CP) was assessed before the behavioral adjustment of the child, reducing the likelihood that behavioral problems of the child evoked CP from the parents. Nonetheless, temporal precedence is not sufficient for establishing a causal relationship. Even though effect sizes found in this study were small, these findings might still be cause for concern at a societal level given that a large proportion of the world’s children experience CP.

An additional problem with the use of correlational longitudinal data is the inability to discern whether a difference found in a construct is due to different initial statuses on a measure, or because of actual rates of growth [21]. For example, aggressive behavior measured in children at two time points might differ between groups because one group actually increased in instances of aggression or because the children in the sample simply entered the study with different initial levels of the construct. To address this limitation, Grogan-Kaylor [21] used a hierarchical linear model to investigate the growth trajectory of CP and externalizing behaviors. In his research, the Behavioral Problems Index (BPI) was administered, in which mothers are asked about the degree to which, for example, a child is hyperactive, aggressive, or breaks rules frequently. Results of this analysis showed that the growth of antisocial behavior over time was linear and that the use of CP had a clear effect on this relationship. Specifically, children who had experienced higher levels of CP exhibited higher levels of antisocial behavior than those who did not receive CP. Interestingly, the author found that the strength of the relationship between CP and antisocial behavior was increasingly strong as children grew older, contrary to the belief that parents might begin to use non-coercive discipline tactics as their children develop. Altogether, this study provides methodologically sound support for the conclusion that harsh physical discipline is associated with negative child outcomes. 

Another common controversy surrounding the use of physical punishment to discipline children is that it does not explain to them why a certain behavior is wrong, nor does it provide alternatives for the child to better handle a challenging situation in the future. In the study by Simons and Wurtele [18], both parents and children were assessed on the practices and beliefs surrounding corporal punishment using vignettes of moral, social, and prudential transgressions. Interestingly, frequent spanking, as reported by the parents, was the strongest predictor of children’s acceptance of aggressive problem solving. Hitting was seen by children as an effective strategy for resolving interpersonal conflict, as indicated by children’s responses to problem-solving vignettes, and they further generalized that hitting was acceptable in most conflict situations toward both peers and siblings. Although data in this study was based solely on self-report measures and may have been affected by social desirability or recall errors, these findings imply an intergenerational transmission of physical punishment, illustrating that CP experience predicts CP acceptance. 

The findings regarding physical discipline and externalizing behaviors provide a methodologically firm conclusion: physical discipline, in both high and low frequency, is associated with adjustment and behavioral problems, antisocial behavior, and endorsement of physical punishment as an appropriate problem-solving strategy. Clearly, to best support children’s development, parenting advisors should encourage parents and caregivers to avoid CP as a disciplinary method.

#### Bidirectionality between Physical Discipline and Externalizing Behaviors in Children

Currently, little is known about the bidirectionality of the relationship between harsh physical discipline and adjustment problems in children; does parental use of physical discipline make children more aggressive, hyperactive, and oppositional, or do these behaviors elicit more physical discipline from parents [22]? A large body of research supports the hypothesis that physical discipline is related to externalizing behaviors. However, as mentioned previously, a large majority of this research is correlational, and very little is known about the directionality of the relationship [15]. Unsurprisingly, parenting literature indicates that there are unique contributions from both the parent and the child that shape their relationship and interactions; a child’s behavior can evoke certain reactions from a parent or caregiver, which in turn evokes specific parenting behaviors [23]. Further, as parents and children develop over time, it is plausible that the nature of this bidirectional relationship can change. As a result, it is important to examine how the parent-child relationship exists and evolves through development to optimize the timing and type of intervention before adverse effects of physical discipline are exacerbated [22].

In their longitudinal study, Verhoeven et al. [22] investigated the possibility of a bidirectional relationship between parenting styles and young boys’ externalizing behavior. At four stages throughout their childhood (17, 23, 29, and 35 months of age), mothers reported a broad range of parenting dimensions, including parental support, structure, and physical discipline using questionnaires. The child’s aggressive behaviors and attentional problems were reported using the CBCL. Interestingly, no evidence of bidirectionality was found, suggesting that children’s behavioral problems influenced parenting behaviors; the boys who displayed high levels of externalizing behaviors were the ones who evoked harsh punishment from their mothers and fathers. The authors explain this finding by suggesting that when dealing with a difficult child, parents may lose control and turn more quickly to harsh punishment tactics. Further, the authors posit that the developmental period chosen in the study may have been too early for parenting behavior to display influence on aggressive behavior or attentional problems. Lansford et al. [15] addressed this limitation by assessing how the relationship between physical discipline and externalizing behaviors develops over the course of time from middle school into adolescence. Mothers reported their discipline practices annually during home interviews, while children’s externalizing behavior was measured using the CBCL administered to the child’s teacher, reducing the likelihood of common method bias. Consistently high levels of externalizing behaviors between ages six and nine were found, where more frequent physical discipline was associated with more frequent externalizing behaviors at all ages. These externalizing behaviors, in turn, were associated with subsequent mild and harsh physical punishment. In the second study, this time consisting of a sample of adolescents aged 10–15, physical discipline and subsequent antisocial behaviors were positively associated, but antisocial behavior was no longer associated with subsequent physical punishment. These findings indicate some evidence for transactional processes between parents and younger children, but these effects appear to weaken as a child reaches adolescence. 

In the previously mentioned studies, only maternal reports of externalizing behaviors and/or physical discipline were included, overlooking the important role of fathers in the relationship between the two constructs. Unsurprisingly, in two-parent families, children are likely to be spanked by both the mother and father, and the quality of the father-child relationship can have a significant impact on a child’s wellbeing [23]. To address this gap in the literature, Lee et al. [23] examined how instances of spanking by mothers and fathers contributed to child aggression in the first five years of life. Associations between the two constructs were measured when the child was one, three, and five years of age. A father-child transactional process was not supported as spanking by fathers was not associated with subsequent increases in child aggression. Further, no evidence was found that fathers changed their rate of spanking as a result of the child’s prior aggression. Results did indicate that mothers spanked more than fathers and that maternal spanking led to increases in child aggression, suggesting that the mothers may have had more opportunities to reinforce this association. 

In sum, the results of these studies found mixed support for the bidirectional relationship between harsh physical discipline and externalizing behaviors. It is plausible that the characteristics of the sample and measures play a role in determining the direction and strength of the relationship, as highlighted by the fact that evidence for transactional processes was found for younger children and maternal spanking but not for adolescents and father-reported spanking. Clearly, further empirical research is needed to assess the reciprocal effects of parental discipline and externalizing behaviors in children. 

### 3.2. Potential Moderators: Cultural Normativeness and Parental Warmth

#### 3.2.1. Cultural Normativeness

Within the study of parental disciplinary practice, the question remains whether all practices are appropriate for all populations [7]. Various studies have shown that parenting behaviors are differentially related to children’s adjustment depending on context, suggesting that conclusions drawn from literature may not be universally applicable [24]. For example, in a study by Deater-Deckard et al. [25], a significant association was found between physical discipline and externalizing behaviors in European-American children, leading them to more aggressive behaviors in the school setting, but not in African American children, who showed lower physical aggression scores after physical punishment. Thus, it has been hypothesized that cultural normativeness or ethnic group differences may work to moderate these effects, emphasizing the importance of recognizing that different communities may require different forms of parenting to optimally support children’s development [26]. 

Lansford et al. [7] approached the investigation of the role of ethnicity on discipline responses by following a sample of children in the United States from pre-kindergarten to adolescence. Over this period, mothers reported their use of physical discipline at multiple time points, and mothers and adolescents reported a variety of externalizing behaviors at age 16. Their analysis revealed significant interactions between race and physical discipline during the child’s first five years of life in predicting adolescent externalizing outcomes. More specifically, results of their analysis showed that the experience of physical discipline at each time point was related to higher subsequent levels of externalizing problems for European American adolescents, but not for African American adolescents. Adolescent externalizing behaviors included items like getting in trouble with the police, getting into fights, and having difficulties at school. Nonetheless, it is plausible that these interactions have roots other than cultural ones. For example, Pinderhughes et al. [27] found significant relationships between African American ethnicity, harsher discipline responses, and increased instances of stress in parents, with stress accounting for the ethnic differences in the tendency to use physical punishment. Moreover, in the context of the many difficulties and challenges faced by minority groups, the effect of the added stressor of physical punishment may be diminished in the face of greater concerns. 

Children’s perceptions of normativeness play an important role in the interpretation of their parents’ disciplinary tactics, as a child who accepts their parents’ behavior as “normal” may not react as strongly when receiving physical punishment. Parents who believe they are acting in a normative way are less likely to act out in uncontrollable anger, and subsequently the message transmitted to the child may not be that their parent is unpredictable and out of control, but instead that the punishment, although unpleasant, is for their own good [24]. Lansford et al. [24] examined these effects by conducting interviews with a sample of 336 mother-child dyads from China, India, Italy, Kenya, The Philippines, and Thailand. Their results showed that children’s perceptions of normativeness of physical discipline (e.g., spanking, slapping, or shaking) moderated the association between physical discipline and child aggression. Specifically, more frequent use of physical discipline was associated less strongly with adverse outcomes (in this case, heightened levels of aggression) when perceived as normative by the children. Using the same international sample, Gershoff et al. [28] found corporal punishment, yelling, and scolding to be associated with child aggression. Interviews were conducted orally in the homes of the participants, and mothers reported how often they used certain discipline techniques, while both mothers and children reported how normal they perceived certain techniques to be in their communities. Measures of the children’s externalizing behaviors were obtained using maternal reports of the aggression subscale of the CBCL. Replicating the results of Lansford et al. [24], perceptions of normativeness were found to moderate the relationship between CP and child aggression. In other words, although more frequent CP was associated with more aggression, the association was less strong when children perceived CP to be normative in their communities. Nonetheless, it is important to keep in mind that translation of interview questions may have led to a loss of information or misunderstanding. Furthermore, findings cannot be generalized to entire cultural groups or subgroups.

The group differences hypothesis explains that patterns of correlations between physical punishment and behavioral problems may differ between cultural contexts. For example, certain behaviors may be adaptive for one cultural group but not another [25]. Polaha et al. [9] examined these differences in a sample of African American and European American children and their mothers. The authors used same-source and distinct-source data, postulating that a child could behave differently across contexts, leading to differential perceptions of the child by parents and teachers. The researchers asked mothers of children aged three to five to estimate the frequency of a variety of disciplinary tactics over the past year, as well as hypothetical questions probing disciplinary responses (for example, *“What would you do if your child got angry at you and hit you?”*), allowing parents to endorse physical and nonphysical strategies. On the other hand, the mother and the child’s preschool or daycare teacher reported measures of child externalizing behavior. The results of their research indicate, as in the previously mentioned studies, that mother-reported physical discipline was associated with more mother-reported externalizing behaviors in children at preschool age. In addition, the authors found a significant two-way interaction with ethnicity, but only when predicting teacher-reported behavioral problems. Specifically, a negative correlation between mother-reported physical discipline and teacher-rated externalizing behaviors was found, but only for African American children. Therefore, using same-source and distinct-source data may contribute unique results to the parenting literature. 

To sum up, some evidence for the moderation by cultural normativeness was found. Cultural normativeness may work to buffer some of the effects of harsh physical discipline, although the exact mechanisms of this relationship did not provide a firm conclusion. Nonetheless, it appears there are culturally specific reasons for, and responses to, physical discipline which may work to buffer the adverse effects of these practices. Clarifying the role of cultural differences in parental discipline is vital in preventing the development of children with disruptive and aggressive traits into problematic adults. 

#### 3.2.2. Parental Warmth

Over the past few decades, a powerful moderator of the correlation between physical punishment and child behavioral problems has been indicated in studies involving a range of research methods [12]. Parental warmth, characterized by affection, comfort, and concern, is thought to promote reciprocity and respect between parent and child, leading to fewer disruptive child behaviors [29]. Physical punishment and parental warmth have been found to be able to co-occur in families, and thus warmth has been hypothesized to act as a buffer against the negative effects of physical discipline as it promotes a positive parent-child relationship [16]; hence, in a setting where a child misbehaves and receives physical discipline from a mother or father who is generally warm and supportive, a child may be less inclined to perceive the parenting environment to be rejecting, protecting them from acting out [12]. 

In their research, Deater-Deckard et al. [12] hypothesized that the correlation between physical punishment and externalizing behaviors would be highest when maternal warmth was lower in a sample of children between one and nine years old. Families completed interviews, surveys, and home observations probing the harshness of discipline and positive feelings about the child, as well as the child’s externalizing problems. As hypothesized, the moderator effect was present; among the lower warmth mother-child dyads, harsher discipline was positively associated with externalizing behaviors. However, the size of the interaction effect found in this study was small and interview data was cross-sectional. In contrast, Lee et al. [16] examined the longitudinal associations between maternal spanking and child aggression in a sample of mothers and their children. Between the ages of one and five, mothers reported instances of spanking and their child’s aggressive behaviors, and maternal warmth was consistently observed. Their results indicate that child aggression was highest in groups characterized by low warmth and that increases in spanking predicted increases in child aggression. However, spanking was unrelated to maternal warmth, such that spanking was similarly associated with high levels of child aggression regardless of whether a mother displayed high levels of warmth. Similarly, Mackenzie et al. [30] administered telephone interviews with mothers in the United States and found that high-frequency spanking at age three significantly predicted externalizing behavior at age five, as indicated by the aggression and rule-breaking subscales of the CBCL. Corresponding to the findings of Lee et al. [16], no significant moderation of maternal warmth in the relationship between spanking and later externalizing behaviors was found.

In the study of Lansford et al. [29], the research was extended to a diverse set of countries to assess whether and how parental warmth moderates the link between CP and aggression in different contexts. Interviews were conducted with mothers and their children, between the ages of seven and ten, in China, Colombia, Italy, Jordan, Kenya, the Philippines, Thailand, and the United States, with a follow up interview a year later. Mothers were asked how often they used certain CP tactics, and maternal warmth was measured using the Parental Acceptance-Rejection/Control Questionnaire. In this study, both mothers and children completed the CBCL to report externalizing behaviors. Maternal warmth was found to be related to decreases in aggression, while CP predicted subsequent adjustment problems. Significant interaction effects were found in some countries; however, results differed depending on whether mother-reported or child-reported aggression was examined. Lau et al. [26] examined the contextual factors that might affect the link between physical discipline and child behavioral problems in Black and White families, assessing warmth as a potential moderator. In their study, physical discipline was associated with increases in externalizing problems when children displayed behavioral problems at an early age. However, while parental warmth protected against later behavioral problems among White children, warm attitudes intensified early problems in Black children. Thus, it appeared the moderating effect of parental warmth on externalizing behavior was dependent on the cultural context or ethnic group in which it occurred. In contrast, McLoyd and Smith [31] used data from a European American, African American, and Hispanic sample of children and found a significant interaction; maternal emotional support (i.e., warmth) moderated the link between spanking and problem behavior for all three racial-ethnic groups. In their study, mothers reported their use of spanking and their children’s behavioral problems, while maternal emotional support of the children was measured during home observations over the course of six years. Findings showed that spanking was associated with increases in behavioral problems over time, but only in the context of low levels of emotional support. 

Based on the studies mentioned above, there is not yet a firm conclusion about the exact role of parental warmth in moderating the relationship between physical discipline and externalizing behaviors. However, these findings highlight the possibility that even in a trusting, warm, and supportive environment physical discipline may compromise the parent-child bonds and reinforce the behaviors it aims to eliminate.

### 3.3. Harsh Physical Discipline and Young Children’s Educational Outcomes

Several studies have linked the use of corporal punishment in children to worsened school outcomes such as poorer educational achievement, disruptive classroom behaviors, and less satisfying peer relationships [7,8]. Various pathways for these associations have been suggested. Physical punishment, although non-abusive, can be experienced by a child as frightening and stressful, and consistent stress can negatively influence a child’s executive functioning. The executive functions are responsible for major mental skills including reasoning, learning, planning, and self-control, which are abilities a child needs to succeed in the classroom environment [8]. Secondly, as mentioned earlier in this review, physically punishing a child predicts a child’s acceptance of CP as an effective problem-solving strategy, which may compromise their bonds with peers [18]. The following section will review the literature on the effects of harsh physical discipline on young children’s educational outcomes, ranging from their academic performance to the quality of peer relationships.

Font and Cage [8] examined how physical punishment scores in a sample of children and adolescents in the United States were associated with cognitive performance, school outcomes, and peer isolation over the span of three years. The sample of children observed were between the ages of eight and 14 at baseline, and both child and caregiver reported that punishment measures were included. Results revealed that mild and harsh forms of physical punishment were negatively associated with school engagement and peer isolation, indicating that students who were physically punished at home often feel lonely and dissatisfied at school. Additionally, Bodovski and Youn [32] examined the role of family emotional climate and its impact on kindergarten children’s academic achievement and classroom behavior. The researchers followed a sample of kindergarteners through to the end of 8th grade and assessed family climate by asking how often a parent had spanked in the last week and if they would hit back if their child hit them. Mathematics and reading scores on standardized tests were reported when the children were in 5th grade, as well as the students’ internalizing and externalizing behaviors in the classroom. Results showed that parental use of physical discipline was associated with lower 5th grade mathematics achievement. Interestingly, this study showed no statistically significant association between physical discipline and later externalizing behaviors.

Parents can contribute to their children’s academic skills training through various activities, for example by helping children with homework, practicing assignments, or reading books together. Literacy stimulation, for example, is a way to improve a child’s cognitive performance that can help them excel in the classroom [33]. In their study, Gest et al. [33] assessed kindergarten children’s emergent literacy skills, frequencies of shared reading between parent and child, and how these related to parents’ responses to common discipline challenges. Significant negative associations were found between parental spanking and language comprehension skills. While a positive association was present between shared book reading and language comprehension skills, it appeared this association did not extend to parents who used physical punishment in discipline situations. Consequently, encouraging parents to use non-directive reasoning may allow children to reap the full benefits of shared book reading in improving their literacy skills. Nonetheless, these results should be interpreted with caution due to the limited sample size of the study.

Moreover, Sherr et al. [34] examined the link between young children’s exposure to violence and educational outcomes with a sample of young children in South Africa and Malawi. Educational measures included school enrollment, school attendance, and progress (whether the child was in the age-appropriate grade). Harsh discipline practices were caregiver-reported and included frequency of slapping and hitting, ranging from weekly to never. Data were collected once in 2011 and again 12 months later. In this study, although exposure to violence at home was very high, physical violence was not associated with enrollment or attendance. Children who experienced harsh physical punishment were less likely to be in the correct grade for their age, but only at baseline. Finally, Amato and Fowler [35] investigated whether associations existed between harsh physical punishment and adjustment problems, school grades, and behavioral problems in a sample of children between the ages of five and 11. Interviews measuring parental support, harsh punishment, and monitoring were conducted with parents and children. Child behavioral outcomes were measured using the CBCL, and parents provided information regarding their children’s school performance in an interview. Results showed that school performance was negatively associated with harsh punishment, while child adjustment and behavioral problems were positively associated. In other words, when parents used hitting as a response to misbehavior, children were more likely to lose their temper, bully others, get suspended from school, and have lower grade point averages.

In sum, the mentioned studies indicate that the use of harsh physical discipline by parents has potentially detrimental consequences in domains outside the home. More specifically, not only cognitive skills, but attendance, school progress, engagement, and relationships with peers may be adversely affected when a child is regularly exposed to violence at home.

## 4. Discussion

In the current literature review, the associations between harsh physical punishment and the development of externalizing behaviors were investigated. Furthermore, the literature on the role of cultural normativeness and parental warmth in moderating these associations and on the effects of harsh physical discipline on young children’s educational outcomes was critically evaluated. 

The findings of the review can be summarized as follows: overall, strong associations between parental corporal punishment and a range of child behaviors were indicated, including aggressive behaviors, antisocial attitudes, destructive habits, and the endorsement of the use of physical violence to solve problems [17,18,21]. These results can be explained by the social learning perspective, which hypothesizes that physical punishment predicts child aggression as children observe these behaviors from their parents and imitate them [1]. Moreover, the finding that children endorsed physical violence as an effective problem-solving strategy may be a result of the inability of CP to promote the internalization of morals [18]. More specifically, the use of physical discipline does not explain why the child’s behavior was wrong or offer alternative suggestions, hindering them from learning socially acceptable responses to challenging or frustrating situations [1]. 

In the reviewed literature, support for a bidirectional relationship between parenting and externalizing behaviors led to mixed results. Transactional processes were found between mothers and children, but not between fathers and their children or mothers and adolescents [15,22,23]. It is possible, however, that cycles of coercion take other forms than physical discipline at older ages [21]. For instance, Grogan-Kaylor [21] suggests that parents may realize the association between their disciplinary style and their child’s antisocial behavior over time and turn to other disciplinary tactics as a result. Furthermore, in many cases, mothers are still more involved with their children than fathers, suggesting they may have more opportunities to negatively influence their child [23]. 

To answer the second research question, this review examined whether any third variables could change the magnitude or direction of the aforementioned effects. Interestingly, the results indicated that children’s perceptions of normativeness may moderate the associations between CP and behavioral adjustment [28]. Specifically, if a child perceived their physical discipline as normative, it was associated less strongly with adverse child outcomes [24]. Gershoff et al. [28] provide a plausible explanation for these results: when children believe their punishment is within a normal range of what their peers are receiving, they are more likely to internalize their parent’s disciplinary message. Thus, when a child experiences their parent’s treatment as reasonable, discipline is less likely to lead to unwanted behavior. It is important to note, however, that while normativeness may have buffered the negative effects of CP, cultural or ethnic differences existed only in the magnitude of these effects. On the other hand, the link never reversed direction, emphasizing that regardless of perceived normativeness, CP was an ineffective means to correcting unwanted behavior.

Secondly, the findings showed that the exact role of parental warmth as a moderator of the effects of physical discipline on externalizing behaviors is not yet clear. While some studies confirmed that supportive and warm parenting can buffer the impact of CP [12,31], further studies did not confirm these findings [16,30] or found that these effects exist only in specific contexts [26,29]. However, it is possible that physical discipline may simply be more prevalent in lower warmth parent-child dyads, intensifying these children’s risk of developing externalizing problems because they are subjected to physical discipline more frequently [12]. Nonetheless, further research is necessary to examine the exact role of parental warmth in moderating the effects of physical discipline on the development of externalizing behaviors and whether these associations are specific to certain contexts or racial/ethnic groups.

Finally, results of the reviewed literature indicate that children who are physically punished in the home experience a range of difficulties at school. Specifically, results of the studies showed that CP is associated with poorer educational achievement, worsened cognitive performance, and feelings of loneliness at school [8,32]. Furthermore, children exposed to CP were more likely to have lower grades, get into fights frequently, or be expelled from school [35]. To explain these effects, Font and Cage [8] postulate that physical violence may activate the physiological stress response in children, which can inhibit self-regulation and attentiveness, providing a plausible explanation for the impaired performance in cognitive tasks. Finally, it is possible that children observe and imitate the violence they see at home, causing their peers to view them as undesirable for friendship [18].

In addition to the findings mentioned above, it is important to mention that the effects found may be a result of the methodological concerns in parenting research. Firstly, many studies examining links between CP and child behaviors measure both constructs at one point in time and thus are not sufficient to infer causal direction [1]. Secondly, as the subject matter pertains to highly personal and sometimes controversial matters, the reliance on self-report data implicates social desirability biases and problems with retrospective recall, which places the validity of the results into question. Finally, the use of correlational data cannot rule out the possibility that third variables explain the associations found [20]. These limitations must be kept in mind when interpreting the results. 

The current literature review is not without its own limitations. Firstly, given the limited scope of the review, it was not possible to examine all the available research on the topic. For example, there have been other moderators implicated in the association between CP and child behavior, such as gender or socioeconomic status [13]. Secondly, this review used the terms physical discipline, corporal punishment, and physical punishment interchangeably. Similarly, although the focus of the paper was on externalizing behaviors, in some cases studies examining general adjustment or behavioral problems were included. While a criterion for the included studies was maintained, it is possible that the authors of the studies defined and measured these constructs differently, leading to differential interpretations of the results. Finally, it is important to mention the cultural context in which this paper exists. As mentioned previously, cultural factors affect the way physical discipline is assessed by a parent or child, and similarly, they affect the way in which this issue is researched. To discuss what forms of discipline are acceptable, we use our own moral standards and cultural norms. Thus, individual views about physical punishment may influence the way physical discipline is measured or interpreted. For example, it is not surprising that when definitions of physical punishment range from spanking to severe beatings, milder forms of punishment become confounded with more severe forms and are consequently strongly associated with negative child outcomes [36]. In the context of this review, it is important to acknowledge culturally varying customs, and to keep in mind that there are many methods for socializing children to succeed in their respective communities [37]. Despite its caveats, this review provides a valuable overview on the effects of harsh physical discipline on the development of externalizing behaviors in children. This review included a range of developmental periods, both maternal and paternal instances of CP, and cross-cultural samples to provide a comprehensive picture. Finally, by examining the impact of CP on educational outcomes, this review is one of the first to combine the overall implications of harsh punishment and its spillover effects into other domains. 

Future research would benefit from longitudinal research designs, ethnically and socioeconomically heterogeneous samples, as well as the inclusion of multiple measures of physical discipline. It is important that future studies consider the cultural differences in the perception of corporal punishment, both by children and their parents/caretakers. As the moderating role of parental warmth in the relationship between harsh physical punishment and externalizing behaviors did not provide a firm conclusion, this is an interesting starting point for researchers in the field to explore further. Further, this study was limited to articles that discussed the use of parental or caregiver corporal punishment as a disciplinary method. The research evaluated in this review indicated that the use of CP in the home can have adverse effects on the child’s behavior in the educational environment. Given this information, it would be interesting to explore whether the use of CP in the educational context (by teachers or other staff) would affect a child in their home environment in a comparable way, and which factors can act as a buffer to these effects. Finally, future studies should subject other disciplinary tactics to comparable scrutiny to ensure these are not similarly associated with undesirable child outcomes.

## 5. Conclusions

The results support several main conclusions. Firstly, the use of physical punishment as a disciplinary tactic is associated with children’s increased instances of aggression, antisocial behaviors, and other behavioral problems. Secondly, cultural normativeness and the overall context in which CP occurs play a role in the way that physical discipline is related to child adjustment. Finally, a child who is exposed to physical discipline in the home is likely to be adversely affected in an educational context as well. Overall, the results provide a firm, overarching conclusion: physical discipline is harmful, ineffective across age groups and cultural contexts, and an unnecessary means to correcting unwanted behavior. This paper further highlights the importance of ensuring government organizations and educators openly discuss the topic of harsh physical discipline with parents and bring awareness to its negative effects in the educational context as well. This way, parents or caregivers can make informed decisions on the best way to raise their children. Oftentimes, parents use the same disciplinary methods that were used for them growing up, but by highlighting unwanted outcomes there is plenty of opportunity to break harmful cycles. In sum, this paper highlights the importance of advising policy makers, parenting advisors, and healthcare professionals to encourage parents against the use of these damaging practices while providing safe and empathetic alternatives.

## Figures and Tables

**Figure 1 ijerph-19-14385-f001:**
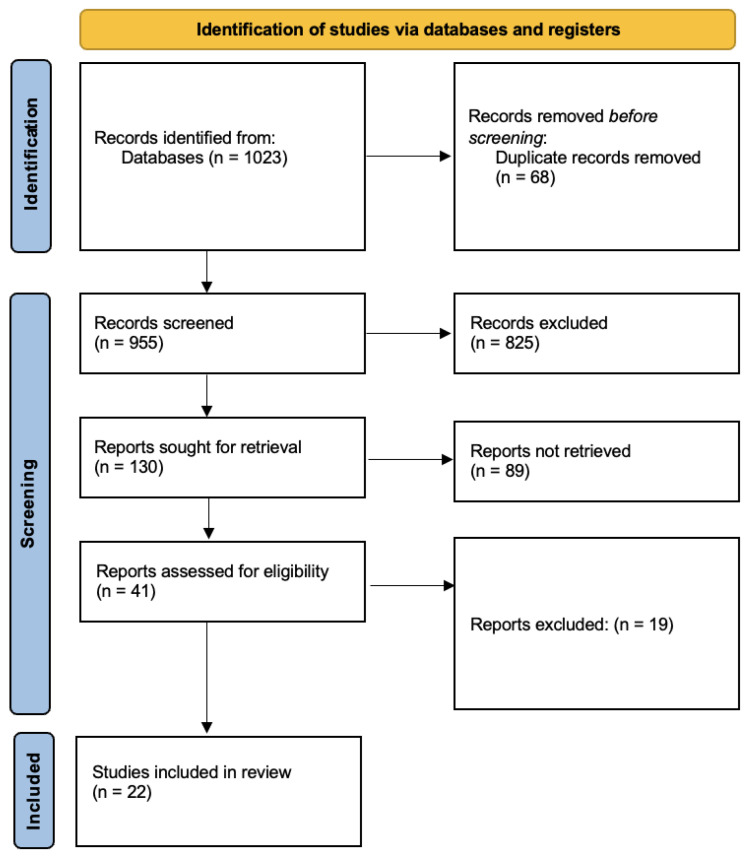
PRISMA flow diagram [14].

**Table 1 ijerph-19-14385-t001:** Overview of studies.

Author	Variables	Nationality
Mackenzie et al. (2012)	Spanking on ext. behavior	Canada
Mulvaney and Merbet (2007)	CP on children’s behavioral problems	United States
Grogan-Kaylor (2005)	CP on antisocial behavior	United States
Simons and Wurtele (2010)	Spanking on endorsement of hitting	United States
Verhoeven et al. (2010)	Bidirectional effects ext. behavior and PD	The Netherlands
Lansford et al. (2011)	Bidirectional effects ext. behavior on PD	United States
Lee et al. (2015)	Bidirectional effects aggr. and PD (father)	United States
Lansford et al. (2004)	Race	United States
Lansford et al. (2005)	Cultural normativeness	United States
Gershoff et al. (2010)	Cultural normativeness	United States
Polaha et al. (2004)	Ethnic differences PD and ext. behavior	United States
Deater-Deckard et al. (2006)	Maternal warmth	United States
Lee et al. (2013)	Maternal warmth	United States
Mackenzie et al. (2012)	Maternal warmth	United States
Lansford et al. (2014)	Maternal warmth	United States
Lau et al. (2006)	Maternal warmth	United States
McLoyd and Smith (2002)	Emotional support	United States
Font and Cage (2018)	PD on school adjustment and cognitive performance	United States
Bodovski and Youn (2010)	Family climate on school achievement	United States
Gest et al. (2004)	PD and reading/language comprehension	United States
Sherr et al. (2015)	Violence on educational outcomes	United Kingdom
Amato and Fowler (2002)	Parenting practices on child adjustment	United States

Note. CP = corporal punishment, PD = physical discipline, ext. = externalizing, aggr. = aggression.

## Data Availability

Not applicable.

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
