# Peer review of "Harsh Physical Discipline and Externalizing Behaviors in Children: A Systematic Review"

_ijerph, 2022, doi:10.3390/ijerph192114385_

Round 1

Reviewer 1 Report

 have carefully read the paper submitted for my attention. I thank the Editor and the authors for their work. The paper is interesting and addresses a crucial topic in the literature, on which research is still needed. The paper appears comprehensive, fluent and well-organized. In my opinion, the manuscript could be accepted in its current form.

I offer some insights, but feel free for the authors and editor to incorporate them in any revision:

1) the literature review in my opinion could be considered a scoping review or even almost a systematic review. I believe that by giving such a slant the work increases its attractiveness.
2) I suggest that the authors provide a summary table of the data collected, indicating authors, year, nationality and any variables of interest for each article included in the review.
3) I suggest the authors extend the considerations to future research. What directions for researchers in this area? I am interested in reading any suggestions and incorporating them into my work.
4) More extensive should be the section on practical implications.

Reviewer 2 Report

Congratulations on taking up a very important and extensive topic. The work presented is well planned and methodologically performed.

However, I have minor reservations about the introduction. In my opinion, it is written very biased. In your work, you point out the cultural differences that affect the perception of corporal punishment by children. I understand this because it is a common punishment in my country of origin. The introduction only contains information on the negative consequences of using this type of educational method. I propose to include information on cultural differences in the assessment of corporal punishment in the introductory part. The topic is very important, and presenting such a strong one-sided narrative at the very beginning may discourage readers from some societies from exploring it further.
